# BENCHMARK DATASET GENERATION AND EVALUATION FOR EXCEL FORMULA REPAIR WITH LLMS

## ABSTRACT

Excel is a pervasive yet often complex tool, particularly for novice users, where runtime errors arising from logical mistakes or misinterpretations of functions pose a significant challenge. While large language models (LLMs) offer promising assistance by explaining formula errors, the automated correction of these semantic runtime errors remains an open problem. A primary challenge to advancing models for such scenarios is the severe lack of high-quality, comprehensive datasets for training and rigorous evaluation. This paper addresses this gap by introducing a novel approach for constructing a benchmark dataset specifically designed for Excel formula repair. We propose a data generation pipeline, which leverages a small set of curated seed samples from online forums to synthetically expand the dataset. Our pipeline integrates few-shot prompting with LLMs and employs a robust *LLM-as-a-Judge* validation framework, combined with execution-based checks to ensure the correctness and semantic fidelity of the generated data. This process produced a benchmark dataset of 618 high-quality samples, covering common runtime errors. Furthermore, we propose a context-aware baseline technique for Excel formula repair that utilizes LLMs to leverage both the faulty formula, and relevant spreadsheet context. We evaluate the performance of various LLMs (GPT-4o, GPT-4.1, Phi-3, Mistral) on our newly generated benchmark using execution-based metrics. Our analysis demonstrates the dataset's quality through manual annotation and provides insights into error and function distributions. The proposed generation methodology is highly scalable and can be readily adapted to create evaluation benchmarks for similar code repair tasks in other low-resource programming languages.

Synthetic Data Generation, Large Language Models, Formula Repair

## 1 INTRODUCTION

Spreadsheets (e.g., Microsoft Excel, Google Sheets) are among the most widely used end-user programming platforms, with hundreds of millions of users worldwide Barowy et al. (2014); Kandel et al. (2011). They empower users—often without formal programming backgrounds—to manipulate and analyze data through formulas composed on a tabular grid. However, writing correct and robust formulas remains a significant challenge. Small mistakes such as incorrect cell references, missing arguments, or improper function nesting can break computations or lead to incorrect results. These errors may surface as syntax problems, logical bugs, or runtime failures (e.g., `#DIV/0!`, `#REF!`), and diagnosing them is often non-trivial for non-programmers Hermans et al. (2016). This highlights the need for tools that assist users in automatically detecting and fixing errors in their formulas.

Automated Program Repair (APR) has been extensively studied for general-purpose programming languages (GPLs) like Java, Python, and C++ Monperrus (2018); Gousios et al. (2015). These systems typically rely on well-structured code with modular functions and comprehensive test suites, which support static analysis and test-driven repair strategies. In contrast, spreadsheet formulas present unique challenges. They operate over structured data objects (e.g., cell ranges, tables) and often lack modular abstractions or formal specifications. Writing correct formulas requires not just syntactic fluency but also a clear understanding of how to reference and manipulate tabular data.

Prior work on APR in spreadsheets (henceforth referred to as *Excel formula repair*) has largely focused on syntactic repair, often without leveraging the spreadsheet context. For example, LaMirage Bavishi et al. (2022a) uses grammar-based candidate generation and a neural ranker to fix syntax errors. A more recent system, RING Bavishi et al. (2022b) applies prompting techniques and error localization to formula text to suggest potential repairs to formula errors. FLAME Joshi et al. (2024), a domain-specific fine-tuned model has shown impressive performance on formula synthesis using compact transformers trained on Excel-specific corpora. However, none of these approaches consider tabular data. Also, most current approaches remain focused on syntax repair, and do not address runtime errors.

Importantly, our analysis of real-world user queries on forums like Stack Overflow and Excel support channels reveals that a majority of formula issues are semantic rather than syntactic. These issues often manifest as runtime errors and require context-aware reasoning over both formula logic and spreadsheet data for effective resolution. Moreover, existing datasets typically include only isolated formula pairs—incorrect and corrected—but omit the spreadsheet context necessary for modeling these semantic repair tasks. There isn't an existing dataset that can be used to train and evaluate a repair model for runtime errors.

**Our Contributions.**    To address this gap, we introduce FOREPBENCH (**Fo**rmula **Rep**air **Bench**mark), a new benchmark and dataset for context-aware Excel formula repair. Our main technical contributions are:

1. **Excel Formula Repair Dataset**: We present FOREPBENCH, the first large-scale dataset of Excel formula repair examples for runtime errors. Each example includes spreadsheet context (cell values, headers), a broken formula, its corrected version, and a user utterance expressing intent. The dataset contains 618 examples and spans 5 runtime error types: #DIV/0!, #N/A, #NAME?, #REF!, and #VALUE!.

2. **Synthetic Data Generation and Validation Pipeline**: We introduce a data generation pipeline that bootstraps from a small number of high-quality samples and produces realistic examples for training and evaluating models for Excel formula repair. We validate each repaired formula both via execution (to confirm correctness) and through a chain-of-thought LLM judge (to ensure semantic alignment with intent), resulting in high-quality examples.

3. **Baseline Approach for Excel Formula Repair**: We propose a method that leverages a large language model and incorporates both formula text and spreadsheet context for error correction. We also report the performance of the baseline approach on FOREPBENCH and seed dataset using various state-of-art proprietary and open-source models.

We have made FOREPBENCH available as a resource for further research on Excel formula repair and related tasks. [1].

## 2    RELATED WORK

### 2.1    LLMS FOR CODE GENERATION

Large Language Models (LLMs) have emerged as a powerful paradigm for code generation. Early work such as GPT-3 Brown et al. (2020) showed that scaling autoregressive transformers in a few-shot setting can yield impressive results in synthesizing code from natural language prompts. This breakthrough spurred the development of models fine-tuned specifically on code, substantially improving both fluency and correctness. OpenAI's Codex Chen et al. (2021a) adapts the GPT architecture with fine-tuning on a massive corpus of public code repositories and supports a wide range of programming tasks—from simple completions to complex algorithmic problems. Salesforce's CodeGen Nijkamp et al. (2022) and Meta AI's InCoder Liu et al. (2022) introduced novel pretraining objectives, including span-masking and infilling, enabling the generation of code that integrates naturally within surrounding context.

Hybrid approaches have also been explored. DeepMind's AlphaCode Li et al. (2022) combines LLMs with search-based techniques, achieving competitive results on programming competitions.

---

[1]`https://anonymous.4open.science/r/FoRepBench-49E7/`

Other models, like CodeT5 Wang et al. (2021), leverage structural information by incorporating representations of abstract syntax trees (ASTs) Yin & Neubig (2018), improving syntactic accuracy and semantic consistency. In our experiments, we evaluate our formula repair approach using four recent LLMs as the backbone: GPT-4o Hurst et al. (2024), GPT-4.1 [2], Phi-3 Abdin et al. (2024), and Mistral Jiang et al. (2023), through prompt engineering and context tailored to the Excel formula repair task.

## 2.2 Excel Formula Generation and Repair

Research on code generation and repair in the context of Excel formulas remains relatively limited. One major line of work focuses on the NL-to-Formula (NL2F) task, which adapts the Text2SQL paradigm to Excel formula generation Zhao et al. (2024). SpreadsheetCoder Chen et al. (2021b) enhances formula prediction by incorporating spreadsheet context, improving the accuracy of generated formulas. FlashFill Gulwani (2011) pioneered example-driven formula synthesis, enabling users to generate formulas via input-output examples. LaMirage Bavishi et al. (2022a) targets the "last-mile" repair problem by fixing near-correct formulas using symbolic and neural techniques, but it does not leverage the surrounding spreadsheet data for deeper semantic reasoning.

Recent work such as FLAME Joshi et al. (2024) introduced a lightweight transformer model for formula completion and repair, trained specifically on Excel formulas. While effective, FLAME operates solely on formula syntax and does not incorporate the spreadsheet context and natural language input, limiting its applicability in user-facing or intent-driven tasks. In contrast, our dataset includes natural language utterances alongside spreadsheet context, faulty formulas, and ground-truth repairs. This enables the study of broader problem settings—ranging from NL-to-formula generation to semantic formula repair—not just syntactic correction or last-mile fixes.

Using LLM-as-a-judge for synthetic data evaluation has emerged as an active research area across text generation Gudibande et al. (2023); Wang & Pavlick (2023); Liu et al. (2023) and code generation tasks Zheng et al. (2023); Chen et al. (2023). Complementary to our work, Singh et al. Singh et al. (2024) proposed an automated method for validating synthetic NL-to-formula datasets using LLMs. Their pipeline classifies and filters low-quality synthetic annotations to improve fine-tuning performance. We adopt a similar idea in our generation pipeline by incorporating LLM-based validation to ensure semantic fidelity in addition to execution correctness in generated examples.

## 3 Methodology

This section describes our proposed methodology for synthetic benchmark generation, which we refer to as BOOTSTRAP GENERATOR. Each data point in the benchmark, which represents a formula repair scenario, must include the following fields (See Figure 2a for an example):

**1. Tabular Data**: The spreadsheet context where the user encountered a runtime error.
**2. Faulty Formula**: A formula that results in a runtime error. In this work, we focus on #N/A, #REF!, #VALUE!, #NAME?, and #DIV/0! errors.
**3. Correct Formula**: A formula that resolves the runtime error and is also consistent with the user intent expressed in the utterance.
**4. Utterance**: A natural language query describing the user's problem and/or task that they are attempting to solve with their formula.

Our data generation method relies on a small set of high quality examples to generate a larger dataset. In Section 3.1, we describe how we curated a set of seed samples. Then in Section 3.2, we describe our synthetic data generation approach which creates FOREPBENCH.

## 3.1 Seed Data Curation

To create a seed dataset for BOOTSTRAP GENERATOR, we developed a systematic approach to collect and process data from online forums where users discuss Excel-related problems and solutions. This section describes the methodology used to gather relevant seed data and prepare it for use in our synthetic data generation pipeline.

---

[2]https://openai.com/index/gpt-4-1/

### 3.1.1 AUTOMATED EXTRACTION AND FILTERING

We scraped posts from the MrExcel[3] forum, a well-established platform where users frequently seek assistance with Excel formulas and share solutions. Next, we filtered posts to ensure that they had all the required elements for our dataset, i.e. a faulty formula, table context, and the correct formula (we used a reply being marked as "accepted answer" on the forum as an indicator). Once relevant posts were identified, we extracted the table context and formulas using parsing techniques for various formats including plain text, code blocks, and specialized markup. Using the extracted data, we reconstructed the structure of the Excel workbooks by identifying different worksheets mentioned in the posts, mapping formulas and values to their corresponding cell addresses, and understanding the relationships between cells, such as which cells are referenced by a formula. We simulated the evaluation of the extracted formulas using Calc.ts[4], an Excel formula evaluation engine capable of interpreting and calculating the results of formulas outside of the Excel application environment. This allowed us to detect the type of errors produced by the faulty formulas and test the corrected formulas to confirm that they resolve the errors. We retained posts for 5 runtime error types: #N/A, #REF!, #VALUE!, #NAME?, and #DIV/0!. For every post that passed, we added one sample to our dataset containing: 1) faulty formula, 2) correct formula, 3) table context, 4) user query, 5) runtime error type, and other metadata. Figure 4 in the Appendix shows an overview of our seed dataset creation approach.

### 3.1.2 MANUAL VERIFICATION AND CORRECTION

Although we applied several automated filtering and validation steps, not all samples met our requirements for high-quality seed samples. This is because 1) a formula that was accepted by a user on the forum as a solution and did not result in a runtime error when executed through Calc.ts could still be *semantically* incorrect, and 2) The table extracted by our scripts could contain data that is not part of the user's intended table context. For example, in one of the samples, a column with the expected outputs was included in the context. A good benchmark sample should test a model's formula repair capability without leaking information.

To address these limitations, we conducted two rounds of reviews to verify and correct the seed data. In the first round, each sample was assigned to one annotator. The annotator was asked to comment on the following:

1. Does the correct formula meet the requirements expressed in the utterance?
2. Does the faulty formula produce the required error type?
3. Is the table accurately extracted from the post?
4. Is the table consistent with the user utterance?

Only a third of the samples annotated in the first round met all the requirements. We therefore had a second round of annotations to verify the labels in the first round, and if necessary, edit the samples. Each sample was reviewed by at least three annotators. If they agreed that the sample met the above requirements, it was added to the final seed dataset. If not, the sample was edited or removed. Examples of edits include clarifying user utterances, replacing incorrect formulas, and deleting ambiguous/trivial cases. This process ensured that for the samples in the seed dataset, 1) The table contains the necessary information for the repair, and not rows with the desired output that might leak information; and 2) The user utterance clearly indicates the intent.

### 3.2 BOOTSTRAP GENERATION

Automatic extraction of data from forums as discussed in Section 3.1.1 led to incomplete and inaccurate samples, and manual validation and correction is not scalable to a large number of samples. In this section, we introduce our BOOTSTRAP GENERATOR approach, using which we created FOREP-BENCH, a large scale benchmark dataset for Excel formula repair focused on runtime errors. Figure 1 shows an overview of the pipeline. We start with a small number of high-quality samples, i.e. seed samples (See Section 3.1), and generate a larger benchmark dataset synthetically. The development of the synthetic data generation pipeline was guided by the following primary objectives: 1) Verify-

---

[3]https://www.mrexcel.com/

[4]https://www.microsoft.com/en-us/garage/wall-of-fame/calc-ts-in-excel-for-the-web/?msockid=38b38871134a6f5806f59df512676e0c

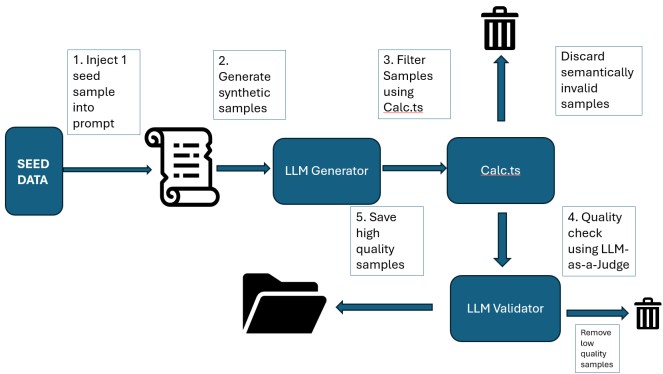

Figure 1: Pipeline overview of the synthetic data generation system.

ing the correct execution of all synthetic formulas in Excel; 2) Ensuring that the data is semantically consistent; and 3) Ensuring appropriate difficulty levels, error coverage, and diversity in functions.

The first step in the pipeline involves generating synthetic samples using few-shot prompting. Following the generation phase, it is essential to validate the synthesized samples to ensure both correctness and quality. This led to the creation of FOREPBENCH, constructed through the following two-stage validation process. The complete pipeline consists of these three steps, as described below.

### 3.2.1 DATA GENERATION WITH FEW-SHOT PROMPTING

We apply *one-shot* prompting to generate synthetic samples utilizing every sample from our seed data. This approach involved injecting each data point from our seed data into a text prompt and subsequently generating new data. This method proved to be the most effective in our validation experiments compared to zero-shot prompting and few-shot prompting with more than one sample. Specifically, 1-shot prompting produced more samples that passed our validation tests (§3.2.2 and §3.2.3) compared to other methods (See §A.2 in the Appendix for more details).

### 3.2.2 VALIDATING GENERATIONS BY EXECUTING EXCEL FORMULAS

To verify correctness, we utilizedCalc.ts[5], to execute Excel formulas against the corresponding spreadsheet data. We ensure that each generated "correct" formula does not result in an error and that faulty formulas produce the appropriate runtime error. After confirming correctness, we evaluate the quality of the generated data using an LLM-as-a-judge framework.

### 3.2.3 VALIDATING GENERATIONS WITH LLM-AS-A-JUDGE APPROACH

To ensure reliability of the generated synthetic data, the LLM-judge approach we implemented leverages Chain-of-Though (CoT) reasoning for systematic assessment. We refer to this model as LLM VALIDATOR. In this work, the LLM-judge analyzes each repaired formula by first determining whether it resolves the original runtime error in the synthetic data. It then assesses whether the formula aligns with the user's intent, considering the spreadsheet context and any provided utterance. It is also prompted to assess and annotate the difficulty level of the repair, which we use for analysis of our proposed benchmark in Section 6.1. Samples that passed both correctness and quality checks are subsequently added to the final dataset.

## 4 FORMULA REPAIR

As discussed in Section 1, there has been scarcity of research done on systems capable of Excel formula repair for formulas that result in *semantic* errors. A key challenge in Excel formula repair lies in incorporating the relevant spreadsheet context. Unlike general-purpose programming languages,

---

[5]https://www.microsoft.com/garage/wall-of-fame/calc-ts-in-excel-for-the-web/

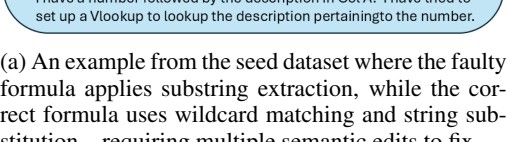

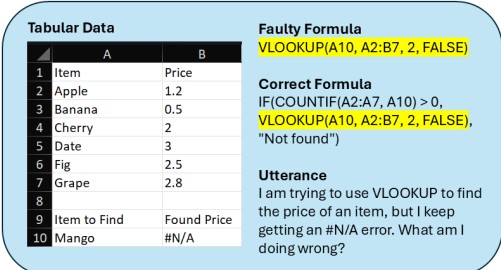

(a) An example from the seed dataset where the faulty formula applies substring extraction, while the correct formula uses wildcard matching and string substitution—requiring multiple semantic edits to fix.

(b) An example from FOREPBENCH. The faulty formula does not handle the exception when the value A10 is not present in the table. The correct formula fixes this by checking if the value exists, and returning "Not Found" if it doesn't.

Figure 2: Example data points contrasting difficulty of the seed dataset and FOREPBENCH

Excel formulas are tightly coupled to tabular data layouts, and the correct repair often depends on values, ranges, headers, or even user-entered text elsewhere in the spreadsheet.

To address this, we propose a baseline solution that not only leverages the erroneous formula and any available auxiliary information (such as natural language descriptions), but also uses the context present within the spreadsheet. Our system thereby enables context-aware formula repair that handles both syntactic and semantic errors. We further utilize our baseline repair pipeline to evaluate FOREPBENCH, thereby demonstrating its practical relevance and showcasing how such controlled benchmarks can effectively approximate real-life formula repair scenarios encountered in production spreadsheets.

## 4.1 BASELINE REPAIR TECHNIQUE

The baseline method follows a structured pipeline that makes a single call to an LLM, designed to efficiently process faulty Excel formulas and generate repaired versions along with explanations (See Figure 1). The system takes as input a faulty formula, the corresponding runtime error, the spreadsheet context, and an optional user utterance. Since spreadsheet tables can be large, passing the entire spreadsheet as context is impractical due to LLM token limitations. To address this, we identify the nearest table associated with the faulty formula and extract its header along with a few sample rows to provide as context. Once the relevant spreadsheet context is retrieved, a structured prompt is constructed, which consists of the extracted spreadsheet data context, the faulty formula, the runtime error, and an optional user utterance; along with task-specific instructions for repairing Excel formulas. The constructed prompt is then passed to the LLM, which processes the input and attempts to generate a corrected formula along with a natural language explanation of the fix.

## 5 EXPERIMENTAL SETUP

We applied BOOTSTRAP GENERATOR(Section 3.2) to generate our dataset using `GPT-4o` as the LLM. A temperature of 0.64 is used to promote diversity among generated examples. In total, we generated 1095 samples, out of which 618 passed LLM VALIDATOR. Table 1 shows the error-wise breakdown. We conducted analysis to assess the characteristics and quality of our dataset. We address the following research questions:

RQ1 Does the data distribution in the FOREPBENCH match real world data?

RQ2 What is the quality of FOREPBENCH based on human judgments?

RQ3 What is the performance of the proposed baseline repair approach across a range of state-of-the-art proprietary ( i.e, `Gpt-4.1`, `Gpt-4o`) and open-source LLMs (i.e, `Phi-3`, `Mistral`) on FOREPBENCH?

RQ4 What is the cost of generating FOREPBENCH? How many LLM calls are needed?

To answer RQ1, we present the difficulty and function distributions for both the seed dataset and FOREPBENCH. The difficulty of each sample was determined by LLM VALIDATOR. It was prompted to assign one of 3 difficulty ratings to the sample - easy, medium, and hard.

To answer RQ2, we had 3 annotators assess the quality of the generated data *before* it was passed through LLM VALIDATOR. Due to the complexity of the task and limited resources, three of the authors with extensive familiarity with the task and deeper understanding of the nuances served as annotators in this study. They annotated a subset of 24 samples from the synthetic dataset. They were asked to perform the same task as LLM VALIDATOR in Section 3.2, i.e. check the correctness and consistency of the table context, faulty formula, correct formula, and utterance.

## 5.1 METRICS

To evaluate the performance of the baseline repair technique across datasets and with different LLMs for RQ3, we employ the following execution-based metrics:

| Metric | Description |
|---|---|
| Syntax Validity | We first check whether the repaired Excel formula can be successfully compiled. If the formula parses without any compilation errors, it is considered syntactically valid; otherwise, it is marked as invalid. |
| Can Execute | This metric verifies whether the repaired formula can be successfully executed on the spreadsheet without triggering any runtime errors (e.g., #VALUE!, #REF!, #DIV/0!, etc.). A successful execution without runtime errors is considered a success; otherwise, it is considered a failure. |
| Execution Match | After execution, we compare the output produced by the repaired formula against the output of the ground-truth (correct) formula. If the outputs match exactly, the repair is considered correct under this metric. |

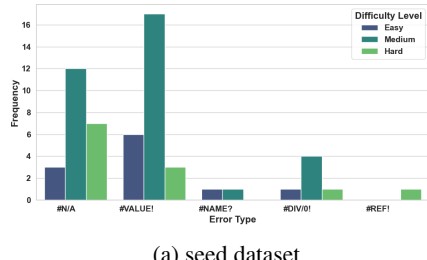

(a) seed dataset

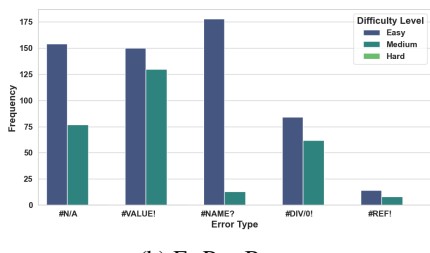

(b) FOREPBENCH

Figure 3: Distribution of sample difficulty levels by Excel error type. Difficulty levels were assigned by LLM VALIDATOR.

| Error Type | Samples Generated | Samples passed by LLM VALIDATOR |
|---|---|---|
| #VALUE! | 241 | 64 |
| #N/A | 329 | 140 |
| #REF! | 16 | 12 |
| #NAME? | 222 | 158 |
| #DIV/0! | 287 | 244 |

Table 1: The table shows the number of generated samples that remained after LLM VALIDATOR filtered out low quality samples. Overall, 56% of all generated samples pass LLM VALIDATOR.

## 6 RESULTS AND DISCUSSION

### 6.1 RQ1: DATA DISTRIBUTION AND COMPARISON WITH SEED DATASET

Figures 3a and 3b display the distribution of difficulty levels across error types in the seed dataset and FOREPBENCH, respectively. Compared to the seed dataset, the samples in FOREPBENCH

| LLM | Dataset Origin | Syntax Valid | Can Execute | Execution Match |
|------|----------------|--------------|-------------|-----------------|
| **GPT-5** | FOREPBENCH | **1.00** | **9.97** | **0.83** |
| | Seed Dataset | **0.99** | **0.69** | **0.43** |
| **GPT-4.1** | FOREPBENCH | **1.00** | 0.96 | 0.80 |
| | Seed Dataset | 0.98 | 0.65 | 0.41 |
| **GPT-4o** | FOREPBENCH | 1.00 | 0.93 | 0.73 |
| | Seed Dataset | 0.96 | 0.63 | 0.35 |
| **Phi-3** | FOREPBENCH | 0.81 | 0.77 | 0.58 |
| | Seed Dataset | 0.73 | 0.41 | 0.24 |
| **Mistral** | FOREPBENCH | 0.78 | 0.76 | 0.51 |
| | Seed Dataset | 0.67 | 0.37 | 0.19 |

Table 2: Performance of Baseline Repair Technique on FOREPBENCH and seed dataset across various LLMs. The Seed Dataset contains 59 samples.

are skewed toward simpler formula repairs. This suggests that BOOTSTRAP GENERATOR tends to produce less complex scenarios, as it's hard to synthesize examples that both execute successfully and pass the LLM-based quality filter, LLM VALIDATOR. The example data points in 2a and 2b illustrate this disparity in complexity. Both examples have faulty formulas with the VLOOKUP function. In the example from the seed dataset in Figure 2a, the user wants to extract a part of a string from a cell, a semantically complex scenario. In a simpler (more syntactic) scenario from FOREPBENCH shown in Figure 2b, the user only needs to handle an exception when the value being looked up isn't present in that cell range.

## 6.2   RQ2: SYNTHETIC DATASET QUALITY BASED ON HUMAN EVALUATION

We calculated pairwise Cohen's Kappa scores for the three human annotators. The results indicate moderate agreement among human annotators (Average Cohen's Kappa: 0.502), suggesting some subjectivity in evaluating the quality of generated samples. Annotators were asked to assess whether the corrected formula appropriately fixes the faulty one and satisfies the intended user operation, and whether the example reflects a realistic spreadsheet scenario. Disagreements often stemmed from differing interpretations of what constitutes a plausible Excel table. For instance, in a sample where a formula attempted to add a string to a number, two annotators interpreted it as a plausible user typo, while the third considered it unrealistic (See Figure 5a in the appendix for more details). This suggests the need for more specific annotation guidelines with detailed examples of diverse scenarios for the annotators to mitigate subjectivity.

We also calculated Cohen's Kappa scores between each annotator and LLM VALIDATOR. Agreement between LLM VALIDATOR and each annotator was also moderate (Average Cohen's Kappa: 0.5). In most cases, LLM VALIDATOR accepted examples that were consistent in formula execution but unrealistic in context. For example,in a sample where the table has textual values like "three" in a numeric column, the LLM VALIDATOR deemed such it valid based on logical consistency, but two human annotators rejected it due to implausible table semantics (See Figure 5b in the Appendix for more details). These results imply that while LLM VALIDATOR is effective at rejecting invalid or illogical formula pairs, it lacks sensitivity to the contextual plausibility of spreadsheet content. As a result, some unrealistic samples may persist in the dataset despite passing automated filtering.

## 6.3   RQ3: PERFORMANCE OF REPAIR TASK ON SYNTHETIC AND SEED DATA

We evaluate the performance of the baseline excel formula repair technique described in Section 4.1 on both FOREPBENCH and the seed dataset. To ensure diversity in our evaluation, we selected four representative LLMs spanning a range of model families, including state-of-the-art proprietary large language models (i.e, `Gpt-4.1`, `Gpt-4o`) and open-weight models (i.e, `Phi-3`, `Mistral`), as well as different architectural paradigms (transformer-based and mixture-of-experts). Table 2 reports the results across the three execution-based metrics described in Section 5.1, evaluated over both datasets using different LLMs that power the baseline repair technique.

Overall, we observe that both the *Execution Match* and *Can Execute* scores are significantly lower across LLMs on the seed dataset compared to FOREPBENCH, indicating that the FOREPBENCH is relatively easier for the repair technique to handle which matches our learning from the RQ1 (See

§ 6.1) on difficulty level distribution. Upon further analysis, we identify two primary reasons for this discrepancy: (1) the faulty formulas in the seed dataset are generally more complex, often involving deeper levels of nesting; and (2) the number of edits required to transform the faulty formula into the correct version is substantially higher.

For instance, consider a faulty formula from the manually annotated seed dataset shown in Figure 2a. This example contains nesting of depth two and uses both VLOOKUP and MID functions. To repair this formula, multiple non-trivial edits are required, including altering the internal logic from $MID(A : A, 8, 15)$ to simply $A : A$ within the VLOOKUP function, followed by additional transformations where the looked-up value is further modified to $B1\&" - "$. Such repairs demand reasoning about user intent and understanding of higher-level semantics, as the modifications involve significant logic changes rather than isolated token-level corrections. In contrast, examples from the synthetic dataset often require fewer edits to achieve the correct formula, as illustrated in Figure 2b. In these cases, the internal logic embedded within the VLOOKUP function typically remains intact, with only minor corrections or additional function wrap-up are needed. Consequently, these repairs are more straightforward for the model to handle, as they involve localized changes rather than substantial semantic rewrites. We also see that bigger GPT-4 series models are better at solving the repair tasks in comparison to Phi-3 and Mistral suggesting more layers and training data can help improve Excel formula repair task.

Overall, these findings suggest that the baseline repair technique is effective on both the seed dataset and FOREPBENCH. In addition, although FOREPBENCH is effective in indicating the relative performance of different models on the task, the BOOTSTRAP GENERATOR pipeline may not fully capture the range of complexity observed in real-world Excel formulas.

### 6.4 RQ4: COST OF DATASET GENERATION

Table 3 summarizes the cost of generating the FOREPBENCH dataset in terms of LLM API usage. For data generation, we issue a single LLM call per prompt to generate $N$ candidate samples (where $N = 25$). From this pool, 1,095 samples contained executable formulas with the correct error type, as automatically verified by Calc.ts. For each of these accepted samples, we issue one additional LLM call to assess semantic validity, resulting in a total of 1,154 LLM calls (59 for generation and 1,095 for validation). This corresponds to an average of approximately 2.09 LLM calls per accepted sample.

On average, each call consumes approximately 2,000 tokens, yielding a per-sample generation cost of approximately \$0.02[6]. These results demonstrate that the BOOTSTRAP GENERATOR pipeline is both scalable and cost-effective for generating large-scale formula repair datasets.

| Generation Calls | Validation Calls | Avg. Calls/Valid Sample |
|---|---|---|
| 59 | 1,095 | 2.09 |

Table 3: LLM call breakdown for generating the FOREPBENCH dataset. Out of 1,095 executable samples, 618 are accepted by LLM VALIDATOR.

## 7 CONCLUSION

In this work, we introduced a modular, low-supervision pipeline for generating and validating synthetic Excel formula repair data, resulting in the FOREPBENCH benchmark. We also proposed a prompt-based repair baseline system to evaluate model performance on both synthetic and real (seed) data. Our investigation across four research questions uncovered key insights about the characteristics, challenges, and utility of the generated data. We show that the synthetic dataset covers a broad range of formula categories and function types and is effective in evaluating the relative performance of different formula repair methods. Increasing complexity of the generated samples and improving the alignment of LLM VALIDATOR with human judgments are future directions we plan to pursue to improve our data generation methodology even further. We hope FOREPBENCH will support future research in end-user programming, intelligent assistants, and robust Excel formula repair.

---

[6]https://llmpricecheck.com/openai/gpt-4o/

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

# A APPENDIX

## A.1 FIGURE ILLUSTRATING THE SEED-DATA CURATION PROCESS

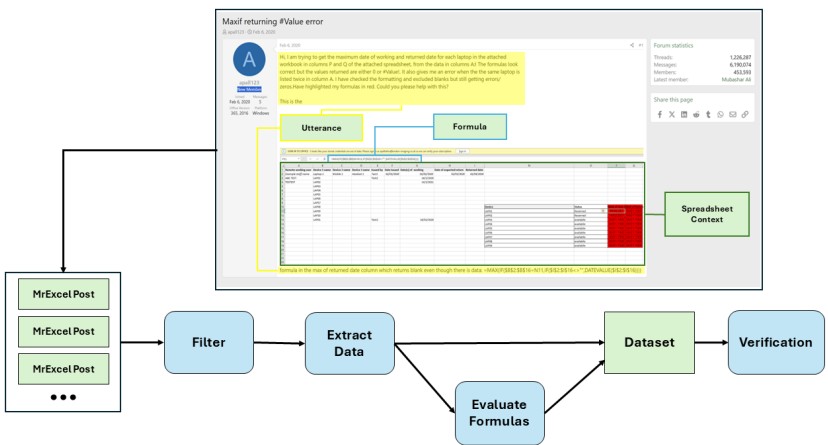

Figure 4: Overall workflow illustrating the sequential steps in the seed data curation process. MrExcel forum is scraped to get posts. Filters are used to identify posts containing table context, faulty formula and correct formula. The page is parsed to extract the required data. The faulty and correct formula are evaluated using Calc.ts. Samples where they execute as expected are added to the dataset. Finally the dataset is manually verified.

## A.2 FEWSHOT-LEARNING SETUP OPTIMIZATION

To generate synthetic samples, we initially employed a zero-shot prompt to evaluate the LLM's capability to generate samples without grounding data. We generated 125 data points per error type, and the key observations were as follows:

1. **Simple Data:** The generated Excel formulas and tables were relatively simple, e.g., dividing an arbitrary cell value by 0 to produce a #DIV/0! error.

2. **Lack of Diversity:** The resulting synthetic data exhibited minimal semantic and syntactic diversity. For a given error type, the model predominantly produced data points with minor differences to the table data or variables in the formula.

Despite extensive prompt modifications to address the aforementioned issues, the problems persisted. Consequently, the next phase involved incorporating real-world grounding data into our prompts as few-shot examples for generating new data.

Next, we explored few-shot prompting as a more promising approach for creating a dataset with more diversity and complex examples. Our validation experiments showed that 1-shot prompting produced more samples that passed our validation tests (§3.2.2 and §3.2.3) compared to zero-shot. Each data point from the seed dataset was used to produce multiple new samples.

## A.3 RQ1 RESULTS: ADDITIONAL DETAILS

Tables 4 and 5 present the distribution of Excel functions in the seed dataset and FOREPBENCH respectively, broken down by error type. We include the top 10 functions that appear in the most examples in this dataset. Comparing the distribution with Table 4, FOREPBENCH frequently has the functions AVERAGE and CONCATENATE which the seed dataset did not. This indicates that BOOTSTRAP GENERATOR creates samples diverse from the fewshot examples. Some trends remain consistent between both datasets, for example VLOOKUP, INDEX, and MATCH being the most common functions for #N/A error.

| Error Type | AND | FIND | IF | INDEX | LEFT | MATCH | MID | MIN | SUM | VLOOKUP |
|---|---|---|---|---|---|---|---|---|---|---|
| #DIV/0! | 0 | 0 | 1 | 0 | 0 | 0 | 0 | 2 | 1 | 0 |
| #N/A | 1 | 0 | 2 | 8 | 2 | 10 | 1 | 0 | 1 | 7 |
| #NAME? | 0 | 0 | 1 | 0 | 1 | 0 | 0 | 0 | 0 | 0 |
| #REF! | 0 | 0 | 0 | 0 | 0 | 0 | 0 | 0 | 0 | 0 |
| #VALUE! | 2 | 4 | 12 | 2 | 2 | 3 | 3 | 1 | 2 | 0 |

Table 4: This table reports the frequency of the top 10 most frequent functions in the seed dataset (N=59), split by error type. Darker color signifies higher frequency within each row.

| Error Type | AVERAGE | AVRG | CONCATENATE | COUNT | IF | INDEX | MATCH | SUM | VALUE | VLOOKUP |
|---|---|---|---|---|---|---|---|---|---|---|
| #DIV/0! | 3 | 0 | 0 | 4 | 1 | 0 | 0 | 9 | 0 | 0 |
| #N/A | 1 | 0 | 0 | 0 | 1 | 25 | 28 | 1 | 0 | 66 |
| #NAME? | 2 | 4 | 0 | 0 | 2 | 0 | 0 | 1 | 0 | 0 |
| #REF! | 0 | 0 | 0 | 0 | 0 | 0 | 0 | 1 | 0 | 2 |
| #VALUE! | 9 | 0 | 16 | 0 | 10 | 2 | 3 | 11 | 4 | 0 |

Table 5: The table reports the frequency of the top 10 most frequent functions present in FOREP-BENCH (N=618), split by error type. Darker color signifies higher frequency within each row. Comparing the distribution with Table 4, FOREPBENCH frequently has the functions AVERAGE and CONCATENATE which the seed dataset did not. This indicates that BOOTSTRAP GENERATOR creates samples diverse from the fewshot examples. Some trends remain consistent between both datasets, for example VLOOKUP, INDEX, and MATCH being the most common functions for #N/A error.

## A.4 RQ2 RESULTS: ADDITIONAL DETAILS

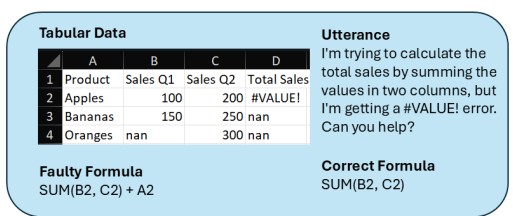

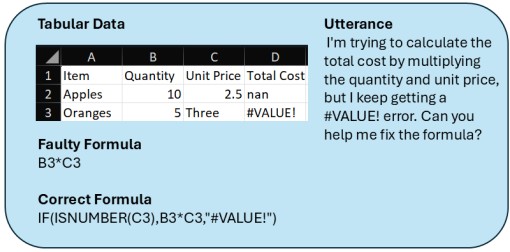

(a) Example from the synthetic dataset *before* being passed through LLM VALIDATOR. Annotators disagree on validity: the formula includes cell A2 in a summation, but A2 contains a string. Two annotators consider it valid as a possible typo, but the third marks it as invalid.

(b) Example from FOREPBENCH where annotators disagree with LLM VALIDATOR on validity. "Unit Price" contains "Three" instead of a numeric value; 2 annotators label it invalid, but LLM VALIDATOR incorrectly accepts it.

Figure 5: Examples to illustrate disagreement between human-human and human-LLM Validator pairs

