# OpenReview forum: "Benchmark Dataset Generation and Evaluation for Excel Formula Repair with LLMs"
_ICLR.cc/2026/Conference — ICLR 2026 Conference Desk Rejected Submission_

### Official Review · Reviewer_muPL · 2025-10-21

**Soundness:** 3
**Presentation:** 3
**Contribution:** 2
**Rating:** 2
**Confidence:** 3

**Summary:**

This paper addresses the critical gap in datasets and methods for repairing semantic runtime errors in Excel formulas, a common pain point for novice users lacking robust APR tools tailored to spreadsheets' unique tabular context. The authors introduce FOREPBENCH, the first large-scale benchmark dataset with 618 high-quality examples spanning five error types (#DIV/0!, #N/A, #NAME?, #REF!, #VALUE!), each including faulty/corrected formulas, spreadsheet context (e.g., cell values and headers), and user utterances. To construct it scalably, they propose BOOTSTRAP GENERATOR: a pipeline starting from 50 manually curated seed samples scraped from forums like MrExcel, expanded via one-shot LLM prompting (using GPT-4o), and rigorously validated through execution-based checks with Calc.ts and an LLM-as-judge framework employing chain-of-thought reasoning for semantic fidelity and difficulty annotation. They also present a context-aware baseline repair technique that prompts LLMs (e.g., GPT-4o, GPT-4.1, Phi-3, Mistral) with erroneous formulas, errors, and extracted table snippets, achieving moderate success on FOREPBENCH as evaluated via execution metrics. Overall, the work demonstrates strong dataset quality through human annotations and error/function distributions, offering a scalable methodology adaptable to other low-resource code repair tasks while highlighting LLMs' potential and limitations in intent-driven formula fixes.

**Strengths:**

- It is an interesting topic and promises to contribute to this area.

- Synthetic dataset is an important research direction.

- The method is straightforward to follow.

**Weaknesses:**

- While the proposed synthetic data generation pipeline is creative, it raises concerns about potential bias and limited generalization. Starting from only 50 manually curated seed samples—sourced exclusively from a single forum (MrExcel), which is both small and unrepresentative—and expanding them via one-shot prompting (GPT-4o, temperature 0.64) risks severe mode collapse and hallucination. In such settings, the LLM is likely to replicate seed patterns mechanically rather than capture the true diversity of real-world spreadsheet errors. The validation process further relies heavily on LLM-as-judge (also GPT-4o), despite recent studies showing that such evaluators can be unreliable, often overestimating quality due to shallow pattern matching. Additionally, the Calc.ts check only verifies syntax and runtime correctness, overlooking edge cases involving intent alignment (e.g., ambiguous user instructions or complex table dependencies). Without incorporating a larger and more diverse seed set (e.g., from Stack Overflow or Reddit) or conducting large-scale human annotation beyond the initial seeds, the resulting FOREPBENCH dataset of 618 samples risks being more of an LLM-driven artifact than a robust benchmark, limiting its applicability in real-world scenarios.

- The evaluation design appears overly simplistic and lacks depth. The chosen baseline (single-shot LLM prompting with minimal context) functions more as a strawman, falling far short of existing systems such as FLAME (2024) or neuro-symbolic approaches like LaMirage extensions, which integrate syntax constraints and search. The study reports only pass@1 execution rates on proprietary (GPT-4o/4.1) and open-source models (Phi-3, Mistral), without any ablation studies (e.g., impact of context extraction, prompt variations, or multi-turn repair), human baselines, or cross-dataset transfer (e.g., SpreadsheetCoder corpus), which limits the credibility of the findings. The “real-world alignment” (RQ1) analysis is largely qualitative, relying solely on forum distribution without quantitative measures (e.g., KL divergence over error types or function usage). Difficulty annotations further depend on unverified LLM judgments, which may diverge significantly from human assessments.

- The scope of the benchmark is narrow, limiting its practical impact. It focuses exclusively on five runtime errors (#DIV/0!, #N/A, #NAME?, #REF!, #VALUE!), while ignoring syntax errors, logical bugs, and multi-formula interactions, according to the authors’ own analysis, which dominate real user queries. The dataset further omits critical spreadsheet features such as multi-sheet references, dynamic arrays (e.g., modern Excel’s SPILL errors), and differences between Excel and Google Sheets, which severely undermines its utility. Claims of scalability also lack substance: there is no detailed cost breakdown (only a vague mention of ~1K LLM calls), no efficiency comparison against rule-augmented approaches, and no evidence supporting adaptability to low-resource languages.

**Questions:**

Please refer to the weaknesses.

---

> ### Author Response · Authors · 2025-11-21
>
> We thank the reviewer for their feedback. We address the concerns raised in the Weaknesses section below, responding to each point separately.
>
> Response to Point 1:
>
> •	In the paper, we discuss multiple prompting configurations used for data generation, including zero-shot, one-shot, and few-shot settings. Although prompting with more than one example can potentially generate more complex samples, the validation pass rate for these settings is significantly lower. One-shot prompting yields the highest number of samples that successfully pass both validators (Calc.ts and LLM-as-Judge). These details are provided in Lines 245–254 and in Appendix A.2.
> •	Our diversity analysis shows that the model is capable of generating diverse samples that are not mere replications of the one-shot in-context example. This analysis is presented in Lines 641–665.
> •	We perform a two-step validation process.
> 1.	Executability filtering (Calc.ts): This step filters out all generated samples that raise syntax or runtime errors, ensuring that only well-formed erroneous formulas (our broken-formula candidates) are retained.
> 2.	LLM-as-Judge filtering: The judge evaluates each remaining candidate by considering the input table, broken formula, reference formula, and user utterance. It removes samples that lack realism or exhibit intent misalignment.
> •	We also attempted scraping additional platforms such as StackOverflow and ExcelForum, where we expected to find a large number of formula-related queries. However, we found very few complete and valid samples belonging to the runtime-error categories. Most real-world posts involve syntax errors or broader semantic issues involving intent mismatch. Therefore, the only platform with a sufficient number of usable runtime-error samples was MrExcel, which we used for seed construction.
>
>
> Response to Point 2:
>
> Our work introduces the first benchmark generation pipeline specifically for semantic, runtime (pound) formula-error repair. To the best of our knowledge, no open-source runtime-error benchmarks currently exist. Moreover, no standard pipeline has been proposed in the literature for generating or repairing runtime errors in spreadsheets.
> Existing systems such as FLAME, RING, and LaMirage primarily address syntax-error repair using grammar-constrained or neuro-symbolic approaches and are not directly applicable to the semantic nature of runtime errors. For this reason, we developed a baseline LLM-based repair approach tailored to runtime-error scenarios, which cannot be addressed by neural validators or grammar-based rules alone.
> The central focus of our paper is benchmark generation, not proposing a novel repair pipeline or analyzing variations within it. Studying the effect of different components of the repair pipeline (context extraction, prompt variations, multi-turn repair, etc.) is an interesting future direction and will be explored in an extension that focuses specifically on runtime-error repair.
> Regarding the reviewer’s point on “real-world alignment” (RQ1), we present both qualitative and quantitative comparisons. We show that the distribution of errors (Figure 3a and 3b) and function usage patterns (Tables 4 and 5 in Appendix A.3) in the generated dataset closely align with the distributions observed in the scraped real-world samples.
>
>
> Response to Point 3:
>
> The main contribution of this work is addressing the lack of open-source runtime-error benchmarks for formula repair. Our pipeline is inherently extendable: it can be used to generate benchmarks for syntax errors or other semantic error categories (e.g., intent-mismatch errors). One can also easily adapt the pipeline for Google Sheets by adding the relevant formulas to the seed set and modifying the in-context examples to reflect platform-specific error types. Thus, the pipeline is designed to be both extendable and scalable.
> We provide a detailed cost breakdown in RQ4, which may have been overlooked. On average, generating a single datapoint costs $0.02, making the pipeline cost-efficient—especially compared to hiring annotators, which becomes prohibitive for low-resource languages.
> Additionally, the size of our dataset (618 samples) is not an upper bound. By adjusting the pipeline hyperparameters and continuing the validation loop, one can generate as many scenarios as needed. The low per-sample cost makes this process scalable and practical

---

> > ### Comment · Reviewer_muPL · 2025-11-28
> >
> > Thanks for your responses! I raised my score to 4.
> >
> > The reliance on a small, single-source seed set and heavy dependence on LLM-based validation still raises questions about generalization and robustness. Furthermore, the evaluation design lacks comparative baselines and quantitative measures of alignment.

---

### Official Review · Reviewer_56zX · 2025-10-24

**Soundness:** 2
**Presentation:** 3
**Contribution:** 2
**Rating:** 4
**Confidence:** 3

**Summary:**

The paper presents a new benchmark for Excel formula repair that considers not only executability but also semantic correctness, i.e., alignment with user intent. The authors first collected seed examples from the web and manually verified them. In the second stage, the dataset is further scaled up using a bootstrap generator approach, by prompting an LLM to generate more problems with 1-shot demonstration of seed problems. Evaluation experiments are conducted on GPT models, Phi-3, and Mistral. Results show the synthetic benchmark is easier than seed real-world samples but useful for relative model comparison. The pipeline is cost-efficient and scalable.

**Strengths:**

* The proposed benchmark considers semantic correctness, which is an important aspect but overlooked by previous works in this domain.
* The curation of the seed dataset is sound and rigorous.
* Experiments and analysis for the benchmarking dataset are comprehensive and inspiring.

**Weaknesses:**

* The seed dataset is a more reliable evaluation set, albeit having a small size. In contrast, the quality of the bootstrapped dataset is concerning. To explain, there is an LLM *examiner* who generates the problem along with a reference answer, and an LLM *examinee* who attempts to solve it. If we only think about generating the answer part, with the same underlying model, the *examiner* has no advantage over the *examinee*, except for the 1-shot demonstration (which is ideally not useful for solving the problem). Consequently, without considering randomness in sampling, there is no way for the *examiner* to come up with a problem that is both difficult enough to fail the *examinee* and solvable by the *examiner* itself. That explains why the LLM-generated problems are generally easier than the seed ones. The difficulty of those bootstrapped problems can never go beyond the capability of the LLM itself. It does not make much sense to me to evaluate an LLM on what it already can do.
* Complementary to the above, I'm concerned about the quality of reference solutions generated by LLMs. Executability checks and LLM-as-judge do not guarantee functional correctness. Empirically, when evaluating the same gpt-4o model that was also used for problem generation, the execution match ratio is still well below 100%, indicating that the model is not confident about the answer across different runs. There's no reason to regard the answer generated during the bootstrap stage as more correct than that generated at test time, given that both use the same model and essentially the same context.

**Questions:**

* What are the reasons for focusing on the task of Excel repair rather than NL2Excel generation from just the spreadsheet and a human instruction? The latter one seems more useful.
* The inline citation format looks non-standard by missing parentheses.

---

### Official Review · Reviewer_SEkX · 2025-10-31

**Soundness:** 2
**Presentation:** 3
**Contribution:** 2
**Rating:** 4
**Confidence:** 4

**Summary:**

This paper introduces FOREPBENCH, a practical benchmark dataset for context-aware Excel formula repair, focusing specifically on semantic runtime errors. The authors present a synthetic data generation pipeline that starts from a small set of curated seed samples and uses few-shot prompting with LLMs to expand the dataset.

**Strengths:**

- The paper addresses a significant gap in the literature. While LLMs have shown promise in code generation and repair for general-purpose languages, their application to spreadsheet formula repair has been underexplored.

- The inclusion of context (table data, headers) and user intent (natural language utterance) is crucial for modeling realistic repair scenarios, moving beyond purely syntactic fixes.

**Weaknesses:**

- **Baseline Method Simplicity:** The proposed baseline repair technique, while context-aware, is essentially a single-prompt engineering approach. It doesn't introduce a novel algorithmic or architectural contribution for repair.

- **Scalability Claim:** The paper claims the methodology is "highly scalable." However, the process relies heavily on a manually curated seed set (59 samples after rigorous manual filtering of forum posts). The scalability of the entire pipeline is therefore contingent on the availability of such high-quality seed data.

- **LLM-as-a-Judge Subjectivity and Reliability:** The human evaluation (RQ2) reveals only moderate agreement (Cohen’s Kappa = 0.502) among annotators. This highlights the inherent subjectivity in judging the quality of formula repairs, especially concerning plausibility and intent alignment. If human experts disagree, the reliability of the LLM-as-a-judge becomes a major concern.

- **Error Type Imbalance** Table 1 shows a significant imbalance in the final dataset, with #DIV/0! (244) and #N/A (140) dominating, while #REF! has only 12 samples. While this might reflect real-world prevalence, it could make the benchmark less effective for evaluating models on rarer error types.

**Questions:**

Q1: How does the reliance on a small, manually curated seed set support the claim of a "highly scalable" methodology, and what are the practical limits to scaling this approach?

Q2: Given the moderate inter-annotator agreement on repair quality, how reliable is the LLM-as-a-judge evaluation, and what steps were taken to ensure its judgments align with a consistent standard?

Q3: “GPT-5” only appears in Table 2, is it a typo？

---

> ### Author Response · Authors · 2025-11-21
>
> 1.	Scalability Claim: We acknowledge that the seed data generation step is not itself scalable. However, once the seed dataset is available, our method can produce a large number of synthetic samples at roughly $0.02 per sample, which is significantly cheaper than expert-created data. This makes the approach particularly valuable for low-resource languages where large human-curated datasets are difficult to obtain.
>
> 2. LLM-as-a-Judge Subjectivity and Reliability: Our LLM Validator uses chain-of-thought reasoning to assess sample consistency. For context, in 5 out of 30 human-reviewed cases, the LLM marked examples as consistent when a majority of humans labeled them inconsistent; overall, there were 7 disagreements in total. While it does not match human performance, the LLM judge achieves an F1 score of 67% in identifying correct samples, indicating reasonable but imperfect reliability
>
> 3. Baseline Method Simplicity:
> We do not claim novelty in the repair model architecture. Our contribution lies in introducing the first benchmark-generation pipeline for formula repair targeting semantic runtime (“pound”) errors, a class not addressed in prior work. Our repair pipeline for runtime errors is similarly the first step toward addressing this gap.
>
> 4. Error Type Imbalance:
> The error distribution in the dataset reflects real-world prevalence and is therefore intentionally imbalanced. Broader error coverage would certainly be useful, but this imbalance does not diminish the utility of FoRepBench.
>
> 5. “GPT-5” Appearing Only in Table 2:
> This is not a typo. We evaluated GPT-5 on the formula repair task. We apologize for the omission in the main text.

---

### Official Review · Reviewer_w1ae · 2025-10-31

**Soundness:** 3
**Presentation:** 3
**Contribution:** 3
**Rating:** 6
**Confidence:** 2

**Summary:**

This paper introduces FOREPBENCH, a benchmark dataset for Excel formula repair, addressing the lack of datasets that capture semantic runtime errors along with spreadsheet context. The authors propose a synthetic data generation pipeline that bootstraps from manually verified seed samples scraped from online Excel forums. The pipeline integrates (1) few-shot LLM prompting for data expansion, (2) execution-based validation for correctness, and (3) an LLM-as-a-judge (LLM VALIDATOR) for semantic and difficulty filtering.
The final benchmark comprises 618 validated samples, each containing tabular context, a faulty formula, a repaired formula, and a natural-language utterance. The authors further propose a context-aware baseline repair approach using GPT-4/4o/4.1, Phi-3, and Mistral, evaluated through execution-based metrics: Syntax Validity, Can Execute, and Execution Match. This paper also includes cost analysis, showing the pipeline’s scalability and cost efficiency.

**Strengths:**

- The paper identifies a genuine gap in semantic formula repair for spreadsheets, which differs significantly from syntax-only program repair tasks in structure and context dependency. By focusing on runtime errors and spreadsheet semantics, it opens a practically relevant and novel research direction.

- The combination of manual seed verification, execution validation, and LLM-judge filtering ensures the generated dataset’s correctness and consistency. The detailed curation and verification pipeline provides solid methodological transparency.

- The proposed BOOTSTRAP GENERATOR can potentially be adapted for similar low-resource domains like SQL or low-code environments, offering a modular framework for synthetic data generation under limited supervision.

**Weaknesses:**

- While the dataset fills an important gap, its final size (618 samples) remains small relative to the diversity of real-world Excel usage. Moreover, the samples tend to be simpler than genuine user-generated errors, limiting the dataset’s stress-testing potential.

- Overreliance on GPT-based validation may bias results.
Because both dataset generation and evaluation rely on GPT-4 variants (e.g., GPT-4o as generator and GPT-4/4.1 as baselines), the benchmark may be inadvertently tuned to GPT’s output distribution, reducing its neutrality for broader model evaluation.

- The chosen execution-based metrics (Syntax Validity, Can Execute, Execution Match) are simple binary checks. They do not capture semantic equivalence, partial correctness, or user-intent fidelity, which are central to runtime error repair.

- The paper does not explore failure cases where LLM VALIDATOR passes logically inconsistent or implausible examples (e.g., textual values in numeric columns), even though such issues are mentioned qualitatively.

**Questions:**

See above.

---

> ### Author Response · Authors · 2025-11-21
>
> We thank the reviewer for their detailed and constructive feedback. Below we address each of the concerns raised:
> 1. Dataset size and representativeness: Our goal is to have a representative dataset for runtime errors. While not comprehensive, we demonstrate that our benchmark is sufficiently rich to reasonably differentiate the performance of different models on the formula repair task. Aa point of reference, a related, widely used benchmark, SPREADSHEETBENCH (Ma et al), has a comparable scale (912 samples).
> 2. Potential bias toward GPT-4o: We used the benchmark to evaluate 4 models in addition to GPT-4o: GPT-4.1, GPT-5 Phi3 and Mistral. GPT-4.1 and GPT-5 outperformed GPT-4o on the benchmark, indicating that benchmark does not systematically favor GPT-4o. However, we acknowledge the possibility of inadvertent bias in the dataset. Our seed dataset and dataset generation pipeline can be used with any language model, not just GPT-4o, to generate an evaluation set.
> 3. Choice of Metrics: This work focuses on creating a benchmark specifically for runtime errors, and the metrics are tailored to formula repair for those errors. Can Execute metric measures whether the formula executes without error, directly reflecting robustness to runtime failures.. Execution Match measures whether the output of the formula matches the ground truth, which captures semantic correctness and user-intent fidelity.
> 4. Human - LLM Judge Agreement: For additional context, in 5 out of the 30 examples reviewed by humans, the LLM passed examples that were voted as inconsistent by a majority of humans. In total, there were 7 instances of disagreement between the majority human label and the LLM judge.

---

### Note · Program_Chairs · 2026-01-17
**Submission Desk Rejected by Program Chairs**

The following references in this submission do not refer to real documents and/or have major errors in bibliographic information:

 Fangyu Liu, Yuxian Wu, Yixuan Liu, and et al. Gpteval: Nlg evaluation using gpt-4 as the referencefree evaluator. arXiv preprint arXiv:2305.04648, 2023.
Lei Zheng, Xiaowei Wang, Baoxu Peng, Xin Wang, and Minlie Huang. Judging code generation with large language models: A comparative study. arXiv preprint arXiv:2305.17951, 2023.
Binyuan Chen, Qian Liu, Jinjie Jiang, and et al. Codellm: Evaluating large language models on code generation. arXiv preprint arXiv:2305.14335, 2023.